# One step synthesis of ultrafine PHF@AuNPs nanocomposite and its application in NIR triggered photodynamic therapy

Wenhao Li[1][☯], Bohao Ruan[1][☯], Xinyi Chen[1], Feng Zhou[2], Xiaoyi Sun[iD][1]*, Yuanyuan Lv[1]*

1 Department of Pharmacy, School of Medicine, Hangzhou City University, Hangzhou, Zhejiang, China,
2 Personalized Prescribing Inc., Richmond Hill, Ontario, Canada

☯ These authors contributed equally to this work.
* sunxiaoyi@hzcu.edu.cn (XS); lvyy@hzcu.edu.cn (YL)

## Abstract

Photodynamic therapy (PDT) is a rapid advancing treatment for cancer therapy. The main challenges in PDT include poor absorption in the "tissue optical window" and aggregation tendency of photosensitizers (PS) such as fullerene in aqueous solutions. Herein, we developed a potent nano PS: fullerene hybrid gold nanoparticles (AuNPs) composites which were ultrafine and well-dispersed with a absorption in near infrared (NIR) region. The composites could be facilely prepared by mixing the reducing and capping agent polyhydroxyl fullerene (2 mg/mL) with $HAuCl_4$ (2.425 mM) at equal volume for 2 h. The obtained composites were negatively charged (−26.3 mv) with the particle size of 14.3 nm. A thin layer of fullerene (~1.6 nm) was coated on the AuNPs core. AuNPs in the composites acted as the light collector, absorbing the NIR light and transferring electrons or energy to the fullerene. Consequently, the composites can be efficiently internalized by tumor cells and activated to produce reactive oxygen species (ROS) intracellularly by 808 nm laser. Enhanced PDT efficacy was observed with the $IC_{50}$ value (50 µg/mL) of the light-activated cytotoxicity and a negligible dark toxicity *in vitro*. This research provides new insights and methods for developing NIR light-triggered fullerene@AuNPs in PDT.

## Introduction

Photodynamic therapy (PDT) is a non-invasive strategy employed in cancer therapy due to its safety and lack of resistance [1]. Once a photosensitizer (PS) absorbs the light with specific wavelength, it will be triggered from the ground state up to an excited state, which then convert oxygen molecules into reactive oxygen species (ROS) to induce cell death. An ideal PS needs to be photochemically efficient with near-infrared (NIR) light activation, darkly non-toxic, accumulated in neoplastic cells, soluble and stable in physiological environments. Unfortunately, conventional PS, like porphyrins, chlorins, and methylene blue mostly have short excitation wavelength in

**Data availability statement:** All relevant data are within the paper and its Supporting Information files.

**Funding:** Yuanyuan Lv: Leading Goose R&D Program of Zhejiang (Grant nos. 2024C03230). https://kjt.zj.gov.cn. The funder provide the fund to support manuscript for publication. The funder had no role in study design, data collection and analysis, decision to publish, or preparation of the manuscript. There was no additional external funding received for this study.

**Competing interests:** The authors have declared that no competing interests exist.

the ultraviolet or visible range [2]. Low light penetration into deep tumor sites makes only superficial lesions on the skin or the lining of internal organs or cavities PDT indications. Therefore, different types of sophisticated PS activated by NIR light, such as noble metals [3], transition metal oxides [4], upconversion nanoparticles [5], aggregation-induced emission (AIE) molecules [6], and two-photon excitation (TPE) nanoparticles [7] have been developed. Fullerene ($C_{60}$) has a high triplet yield for generating ROS via type I and type II photophysical mechanisms due to its extended π-conjugation system [8,9]. In addition to its inertness, biocompatibility, and photo-bleaching resistance, $C_{60}$ holds great potential in PDT. However, like most PS, its visible light absorption region and the extremely low solubility restrict its clinical use [10].

Gold nanomaterials are fascinating and promising candidate in the field of nanoscience and biomedicine owing to their versatile and unique properties [11]. It is known that colloidal gold possesses localized surface plasmon resonance (LSPR) property, which means Au can absorb the NIR light, convert it to heat and finally endow themselves distinctive photothermal and photoacoustic activities [12]. Recently, Yan et al. discovered that $C_{60}$ could be excited by NIR light with the aid of gold nanoparticles (AuNPs) because of the electron transfer between AuNPs with SPR effect and "electronic sponge" $C_{60}$ [13]. They used $NaBH_4$ as the reducing agent in the synthesize of AuNPs which attached to the surface of $C_{60}$-$NH_2$ clusters, resulting in a fullerene based hybrid nanoparticle having a particle size of 188 nm and the zeta potential of −7 mV. This study inspired us to develop a facile hybrid system consisted of $C_{60}$ and AuNPs with more negative charge and smaller size for prolonged systemic circulation time and enhanced particle dynamic stability, being like a fullerene – antenna system [14,15] to achieve fullerene illuminated upon NIR light for PDT.

Even though it has been proved that fullerene has strong inherent affinity towards Au owing to the electron transfer interactions, the preparation of fullerene - AuNPs composite is challenging. Different approaches have been established to obtain fullerene aggregation decorated with AuNPs. The irrespective shortcomings of those methods are complicated surface functionalizations, insoluble, large particle sizes and poor dispensability [16–18].

Herein, we established an extremely simple and novel method to prepare the ultrafine and mono-dispersed polyhydroxyl fullerene (PHF) - AuNPs composites. The obtained composites have unique structure with a dense PHF layer on the perimeter of AuNPs. They can be activated to produce ROS and eliminate cancer cells by 808 nm laser overcoming the short excitation wavelength of fullerene (S1 Fig). With functionalization of hydroxyl groups on the PHF and in-depth *in vivo* study, in perspective, PHF@AuNPs can be used as a new NIR light PS for targeted PDT.

## Materials and methods

### Materials

Fullerene ($C_{60}$) (99.9% sublimed) was purchased from MER Corporation (AZ, USA). Auric chloride ($HAuCl_4$), 1,3-diphenylisobenzofuran (DPBF), 2',7'-dichlorodihydrofluorescein diacetate (DCFH-DA), 3-(4,5-dimethylthiazol-2-yl)-2,5-diphenyltetrazolium bromide (MTT), and Saphedex G-25 were from Sigma-Aldrich

(MO, USA). Fetal bovine serum (FBS) was bought from Thermo Fisher Scientific (MA, USA) and cell culture medium high-glucose (DMEM) was from Genom Biotechnology Co. Ltd (Hangzhou, China).

Human lung adenocarcinoma A549 cells were provided by Institute of Biochemistry and Cell Biology, Shanghai Institute for Biological Science, Chinese Academy of Science (Shanghai, China).

## Synthesis and purification of polyhydroxyl fullerene (PHF)

PHF was synthesized using a previously reported method with slight modifications [19]. Basically, 1.2 g fullerene was dissolved in 500 mL toluene. Tetrabutyl ammonium hydroxide (TBAH, 10 mL) was added into the solution and kept stirring for 3 min before adding 30 mL 50% (w/v) NaOH in a dropwise manner. After stirring for 2 h at room temperature (RT) in $N_2$ atmosphere, the flask was laid aside for phase separation for 5 min. Subsequently, the organic phase was decanted after frozen under −20°C for 2 h. 200 mL $H_2O$ was added and the solution was kept stirring at RT for 4 days. Then, the collected solution was freeze-dried and the obtained PHF was washed with anhydrous methanol.

The purification of PHF was performed on a Saphedex G-25 column. The elution was collected by $H_2O$ used as the eluent till the pH of elution turned basic. Purified PHF was finally obtained through lyophilization process (ALPHA2–4 LSC, Christ, GER): pre-frozen at – 40 °C for 8 h, then, the shelf temperature was raised to −20 °C for 18 h, secondary drying was set at 25 °C for 5 h.

## Preparation and characterization of PHF@AuNPs composites

$HAuCl_4$ (2.425 mM) and 2 mg/mL PHF of equal volume were mixed and pH of the mixture was adjusted to ~10 using 1 mol/L NaOH. After stirring at RT for 2 h, equal volume of methanol was added into the reaction. The composites were washed with $H_2O$ and collected by centrifugation ("a" group).

The aqueous solution of PHF@AuNPs with appropriate concentration was dropped on the copper mesh covered with ultra-thin carbon film. After drying in air, transmission electron microscope (TEM, JEM-1400, JEOL, JP) and high resolution transmission electron microscope (HRTEM, JEM-2100, JEOL, JP) was used to observe the morphology and ultrafine structure of PHF@AuNPs. Energy-dispersive X-ray spectroscopy (EDS) analysis was used to determine the chemical composition of PHF@AuNPs cores.

PHF@AuNPs were dispersed in deionized water with measurable particle concentration, and the particle size and ζ-potential were determined by the nano particle size and zeta potential analyzer (Zetasizer Nano ZS-90, Malvern, UK). UV-Vis spectra of PHF and PHF@AuNPs were recorded by the spectrometer (UV-2450, Shimadzu, JP).

## Reactive oxygen species production by PHF@AuNPs

DPBF was applied as an indicator to detect the ROS production *in vitro*. PHF@AuNPs in phosphate buffered solution (PBS) were mixed with DPBF in DMSO to be irradiated at 808 nm (1 W/cm$^2$) for 0, 5, 10, 15, 20, 25, 30 min (Fiber optic-coupled laser system, Changchun New Industries Optoelectronics Technology Co., Ltd, China). The final concentration of PHF@AuNPs and DPBF were 20 μg/mL and 3 μM, respectively. For the comparison of ROS production, PHF@AuNPs or PHF (20 μg/mL) was mixed with DPBF and treated with NIR laser irradiation or incandescent lamp respectively, as described above. In the multiple irradiation test, PHF@AuNPs were pre-treated with NIR light for 10 min once or twice. Then, DPBF was added. The fluorescence signals were recorded during NIR irradiation for 10 min with emission wavelength at 460 nm and excitation wavelength at 403 nm.

Intracellular ROS production in A549 cells was detected by DCFH-DA. A549 cells were cultured in DMEM medium containing 10% FBS in a humid atmosphere of 37 °C and 5% $CO_2$. $5 \times 10^4$ cells were incubated with PHF or PHF@AuNPs (20 μg/mL) for 4 h and then, the cells were rinsed and irradiated with 808 nm laser (1 W/cm$^2$) for 10 min. The probe DCFH-DA (10 μM) was added for confocal laser scanning microscopy (LSM 980, Zeiss, GER) observation.

## Uptake of and phototoxicity of PHF@AuNPs

The uptake and intracellular distribution study was performed by TEM (JEM-1230, JEOL, JP). Cells were incubated with 20 μg/mL PHF@AuNPs for 4 h, and then washed, fixed, dehydrated, embedded in Spurr resin. Finally, the sections (70–90 nm) were cut (EM UC7, Leica, GER), stained with lead citrate and uranyl acetate, respectively for TEM observation.

PHF@AuNPs, PHF and AuNPs (1 μg/mL, 10 μg/mL, 50 μg/mL) were incubated with A549 cells which were seeded in 96-well plates for 4 h, respectively. Then, cells were washed and irradiated with 808 nm laser (1 W/cm²) for 10 min. For another 12 h incubation, standard MTT assay was performed. The viability of cells incubated with PHF@AuNPs without laser treatment was determined to investigate the cytotoxicity of the nano-composites.

## Results and discussion

### Synthesis and characterization of PHF@AuNPs

Chemical reduction method is widely used for synthesis of metallic nanoparticles. Nuclei will emerge and grow up to the nanoscale when metal ions meet with a reducing agent. Reducing agents especially the weak reducing agents facilitate the formation of small-sized particles with better control of shapes because of the slow reaction rate [20]. $C_{60}$ with weak reducibility is the ideal one for ultrasmall AuNPs production [21]. To improve the hydrophilicity and dispersibility of $C_{60}$, we chose PHF as its derivative to form composites, and the hydroxyl groups on PHF can serve as the active groups to be modified further for targeted therapy (Fig 1).

We investigated the synthesis parameters of reaction time and the ratio of PHF/HAuCl$_4$. After mixing PHF with HAuCl$_4$ ("a" group), the color changed from light pink to wine red at 0.5, 1.2 and 2.0 h. As the reaction time increased, the absorbance of plasmon peak increased and the samples became darker in color (Fig 2A and 2B). When we doubled amount of PHF dosage in the same condition ("b" group, Fig 2A), the composites obtained were virtually the same as those in "a" group. The UV-Vis spectrum showed that all the synthesized PHF@AuNPs had almost the same plasmonic resonance peak at around 520 nm (Fig 2B). Yazdani et al. reported that AuNPs particle size was positively correlated with UV-Vis spectra wavelength peak [22], indicating that the AuNPs we synthesized in this study had similar particle size. Dynamic light scattering (DLS) results confirmed the quite similar hydrodynamic sizes of PHF@AuNPs composites ranging from 10.99～13.17 nm for "a" group and 11.72～13.37 nm for "b" group (S2 Fig). In this chemical reaction, $C_{60}$ enabled the reduction of tetrachloroauric acid to form gold nuclei. Then, these particles grew in size as time increased. A slight size increase was recorded in "a" and "b"group as reaction time increased. However, the change in size of 2–4 nm would

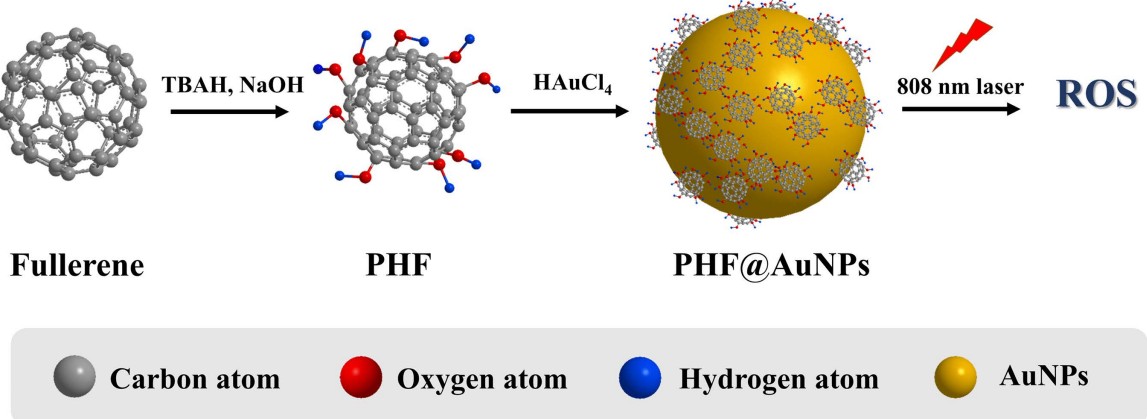

**Fig 1. Synthesis of PHF@AuNPs.**

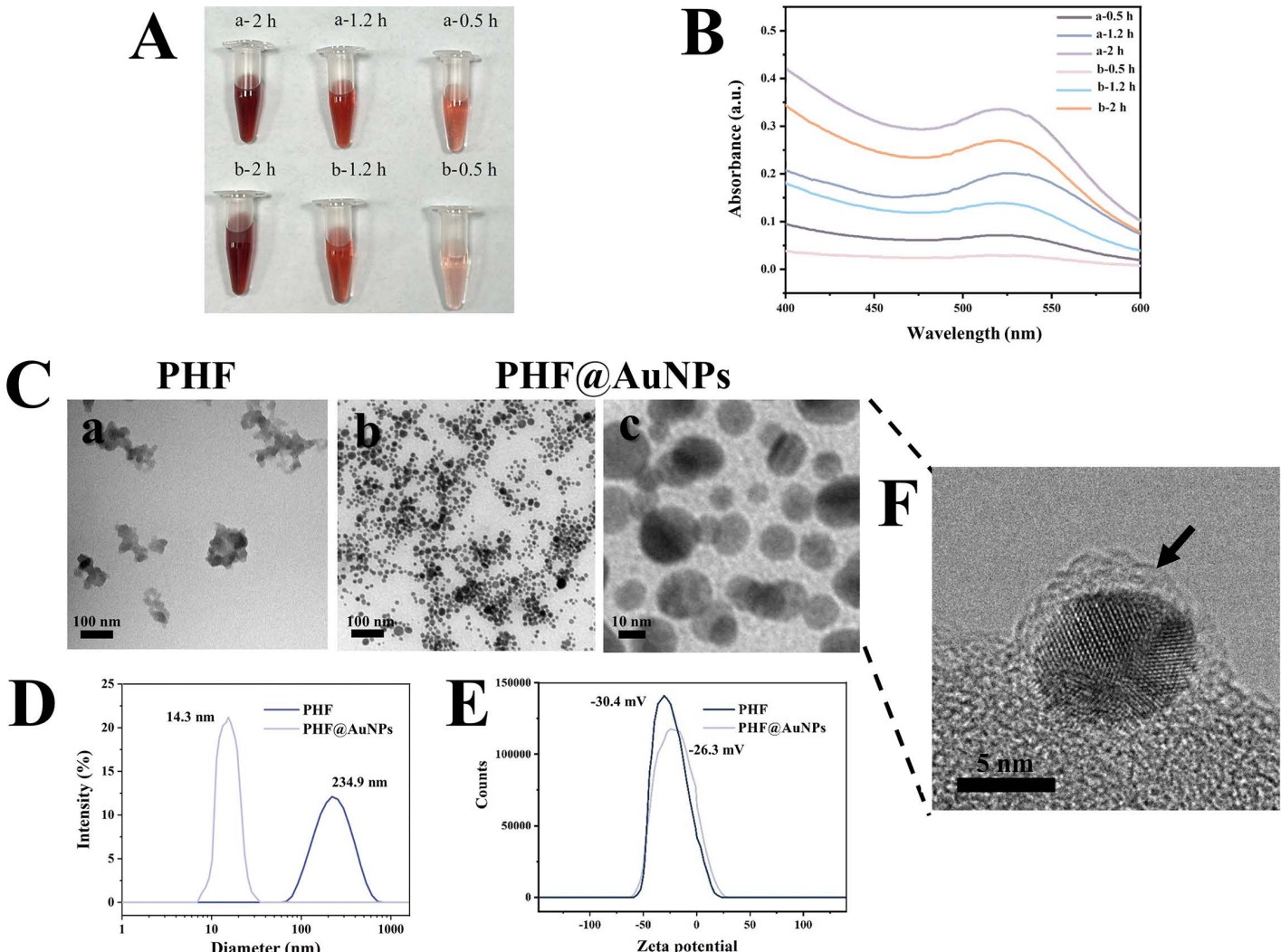

**Fig 2.** (A) The appearance and (b) UV-Vis spectra of PHF@AuNPs in "a" and "b" group obtained at different reaction time. (C) TEM characterization of PHF and PHF@AuNPs in a-2h group (Scale bar: 100 nm in (a) and (b); 10 nm in (c)). (D) DLS results and (E) zeta potential of PHF and PHF@AuNPs. (F) HRTEM image of PHF coating on PHF@AuNPs. The arrow indicates the coating layer of PHF (Scale bar: 5 nm).

not cause SRP shift. During the growth of AuNPs, the density of samples per volume increased, which consequently resulted in an increase in particle yield [22]. We found that once the reaction time exceeded 2 h, the yield would no longer significantly increased. Due to the reduction capability of PHF, even in an extremely short time, the chemical reaction would occur. Only 1 min after mixing, noticeable amount of composites formed, but much inferior in terms of uncontrolled morphology (S3A Fig). When the concentration of PHF was lowered to 1 mg/mL, and concentration of $HAuCl_4$ increased to 10 mM, the morphology and size distribution were much less uniform. The particles exhibited spheres, hexagon, rod morphologies indicating insufficient PHF cannot ensure the growth of AuNPs in a controllable manner (S3B Fig).

Finally, the composites we obtained were generally mono-dispersed spheres (Figs 2C-b and 2C-c). The optimal reaction condition was 2 mg/mL PHF, 2.425 mM $HAuCl_4$ mixing at equal volume for 2 h. The dispersibility of PHF was significantly improved after PHF@AuNPs formation. Compared with the average size of PHF (234.9 nm) measured by DLS, the

particle size significantly decreased to 14.3 nm (Fig 2D). PHF formed aggregations which could be clearly observed in Fig 2C-a. The strong hydrophobicity limits the application of $C_{60}$, although different chemical modifications including hydroxylation have been done to enhance its water solubility of fullerene, significant agglomeration tendency still remained [23]. The commonly used method for control of particle size is liquid-liquid interface precipitation [24]. Obviously, our "mix and go" style reaction is the simplest one.

Meanwhile, stabilizing agents localized at the surface are essential to hinder gold particles growth. The molecule structure of synthesized PHF was $Na_3[C_{60}O_8(OH)_{11}]$ determined by XPS analysis (S4 Fig). The C 1s XPS spectrum shows three peaks at 284.48 eV (C−C), 286.54 eV (C−OH), and 288.17 eV (C=O). Among these peaks, C=O should not exist in PHF. According to Xing's study [25], the "impure groups" (here C=O) are prone to form open cage structure and significantly jeopardize the stability of PHF molecule structure. Here, PHF served not only as an reducing agent, but also an appropriate stabilizer which provided steric hindrance as well as electrostatic repulsion. The introduction of negatively charged hydroxyl group onto the surface of $C_{60}$ significantly augmented its hydrophilicity. The zeta potential of PHF was −30.4 mV. After composites formation, PHF@AuNPs exhibited a slight drop of zeta potential (−26.3 mv) which indicated that a portion of hydroxyl groups on PHF may be shielded (Fig 2E). The high value provided sufficient electrostatic repulsion to allow good physical stability [26].

In order to observe the microstructure of PHF@AuNPs, HRTEM study was further performed. PHF@AuNPs clearly showed an integral thin layer (~ 1.6 nm) of PHF coating on the edge of the AuNPs core (Fig 2F). EDS analysis confirmed the chemical composition of the Au core (S5 Fig). It can be concluded that the layer is derived from PHF. PHF@AuNPs composite successfully formed by using PHF as the reducing agent and capping agent simultaneously.

## Reactive oxygen species production *in vitro*

The SPR effect of AuNPs allows their absorption of NIR radiation, then the excited electrons could transfer to the nearby materials. If the receiver is photosensitizers, the generation of cytotoxic ROS would effectively raise [27,28]. Herein, PHF, as the acceptor, attracted the electrons or electron from NIR irradiated AuNPs in the inner cavity, resulting in the excited state of PHF [14]. DPBF, a ROS probe, its fluorescence intensity is negatively correlated to the ROS yield. Fig 3A depicted the dynamics of ROS production by PHF@AuNPs. The intensity decreased with irradiation time. In order to exhibit the shift of excitation wavelength, the fluorescence intensities of PHF@AuNPs and PHF exposed to 808 nm irradiation were both recorded (Fig 3B). It was clearly showed that PHF@AuNPs were able to release ROS upon NIR light, but PHF wasn't. PHF@AuNPs successfully made redshift of excitation wavelength of PHF. Compared with the initial visible light region, NIR light (808 nm) located in the "biological window (600-900 nm)" which achieves deep tissue penetration and demonstrates more safety. It has been reported that hybrid nanoparticles with a gold layer deposited on $C_{60}$-$NH_2$ clusters enhanced PDT via changing light source [13]. Our findings indicated the redshift of excited light is independent of complex structure. AuNPs and $C_{60}$ are determinants. The comparison of ROS production irradiated by different light source was shown in Fig 3C. The ROS generated by PHF with NIR light was as much as those activated with incandescent lamps. Moreover, the photostablity was confirmed by the multiple irradiations (Fig 3D). PHF@AuNPs could be triggered to the similar level with multiple excitations which holds the advantage over other organic PS.

## Stability of PHF@AuNPs

The stability test of PHF@AuNPs was performed lasting for 1 month. The suspension of PHF@AuNPs in deionized water was stored under 4 °C. The change of particle size, zeta potential and ROS production *in vitro* were determined. It was found that PHF@AuNPs aqueous suspension was physically stable. There was no significant change in size, PDI and surface potential. The ROS production capacity indicated by DPBF rarely decreased (Fig 4).

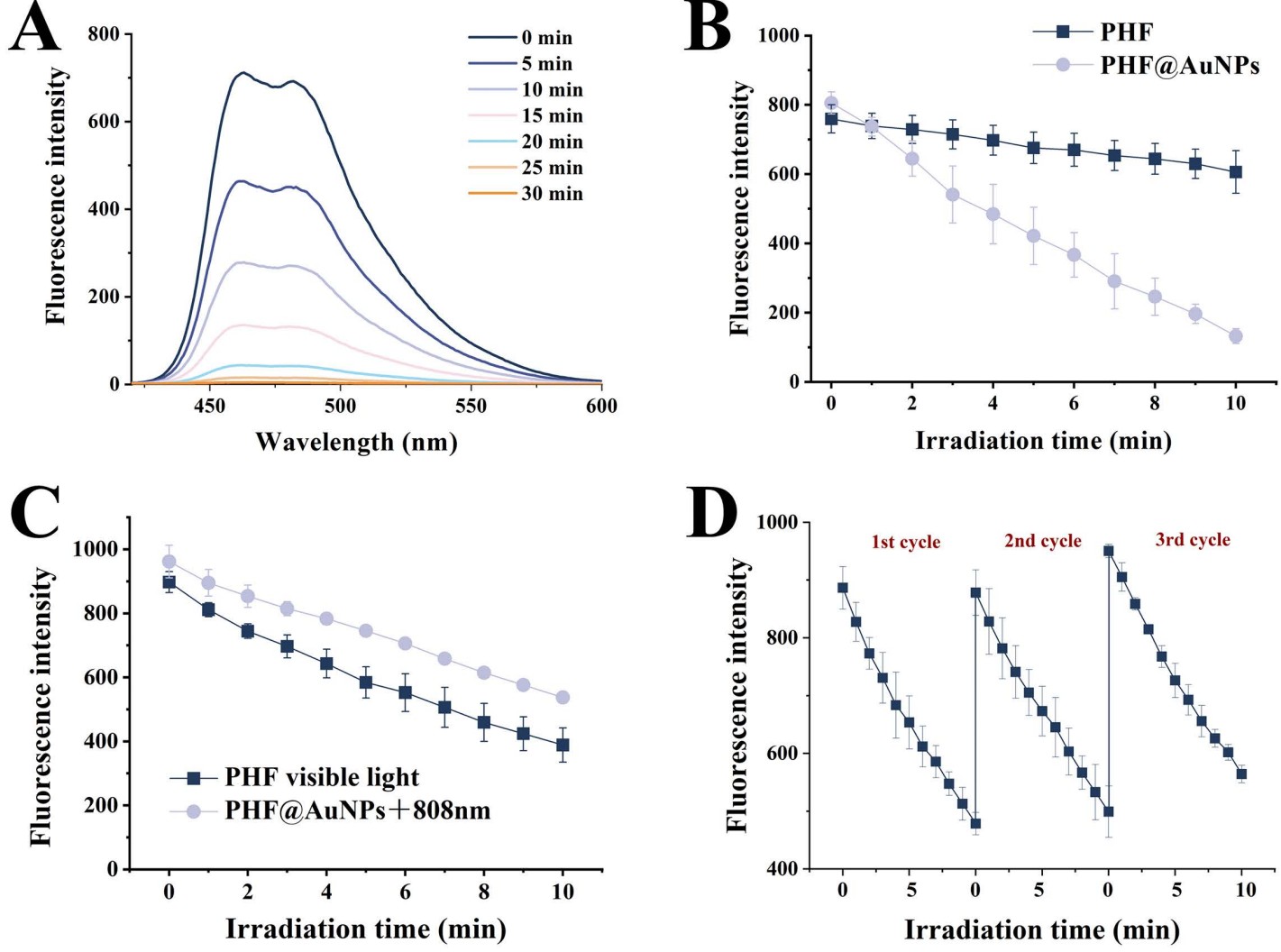

**Fig 3. (A)** The dynamics of fluorescence spectra of DPBF mixed with PHF@AuNPs after exposure to 808 nm irradiation. **(B)** Fluorescence intensity of DPBF mixed with PHF or PHF@AuNPs after exposure to 808 nm irradiation for 10 min (n = 3). **(C)** Fluorescence intensity of DPBF mixed with PHF or PHF@AuNPs after exposure to visible light and 808 nm irradiation, respectively (n = 3). **(D)** ROS generation of PHF@AuNPs during 3 cycles of 808 nm irradiations (n = 3).

## Cellular uptake and ROS detection

Endocytosis is the critical process through which nanoparticles enter into cells. The most researched endocytic routes including clathrin-mediated endocytosis, caveolar endocytosis, and macropinocytosis undergo endo-lysosomal pathways [29]. The results of cell uptake and intracellular location of PHF@AuNPs observed by TEM was consist with the literature report [30]. As displayed in Fig 5, cells endocytosed sufficient composites. They were wrapped and scattered in endo-lysosome vesicles which were the trafficking vesicles in the classical uptake pathway for nanoparticles.

To confirm whether PHF@AuNPs could generate ROS after cell internalization, DCFH-DA, an oxidant sensitive probe was used [13]. After NIR light irradiation, a significant fluorescence was detected in the cells treated with PHF@AuNPs. On the contrary, the signal in PHF group was undetectable (Fig 6). This result was in accordance with that in ROS generation *in vitro*. PHF@AuNPs could produce intracellular ROS with high efficiency by NIR light irradiation.

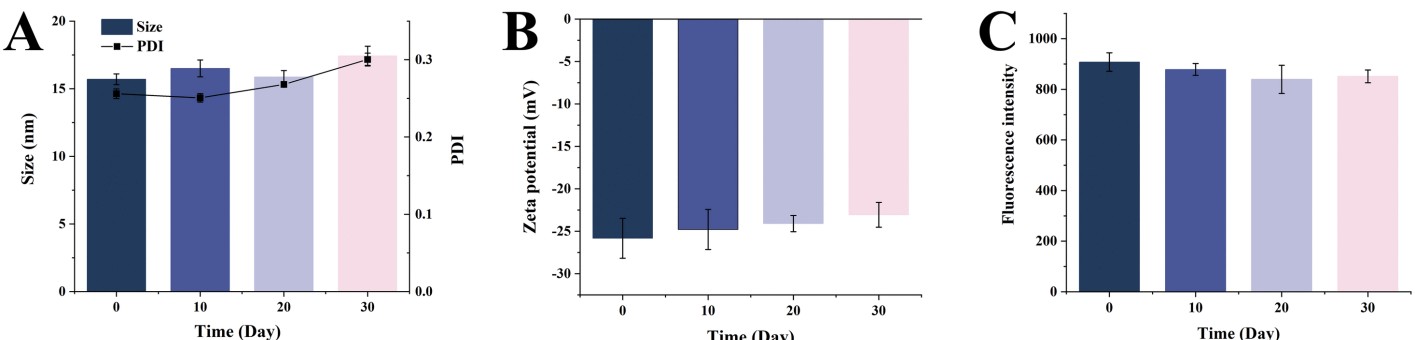

**Fig 4. The change of (A) particle size and PDI; (B) zeta potential; (C) ROS generation capability determined by DPBF of PHF@AuNPs over 1 month storage under 4 °C (n = 3).**

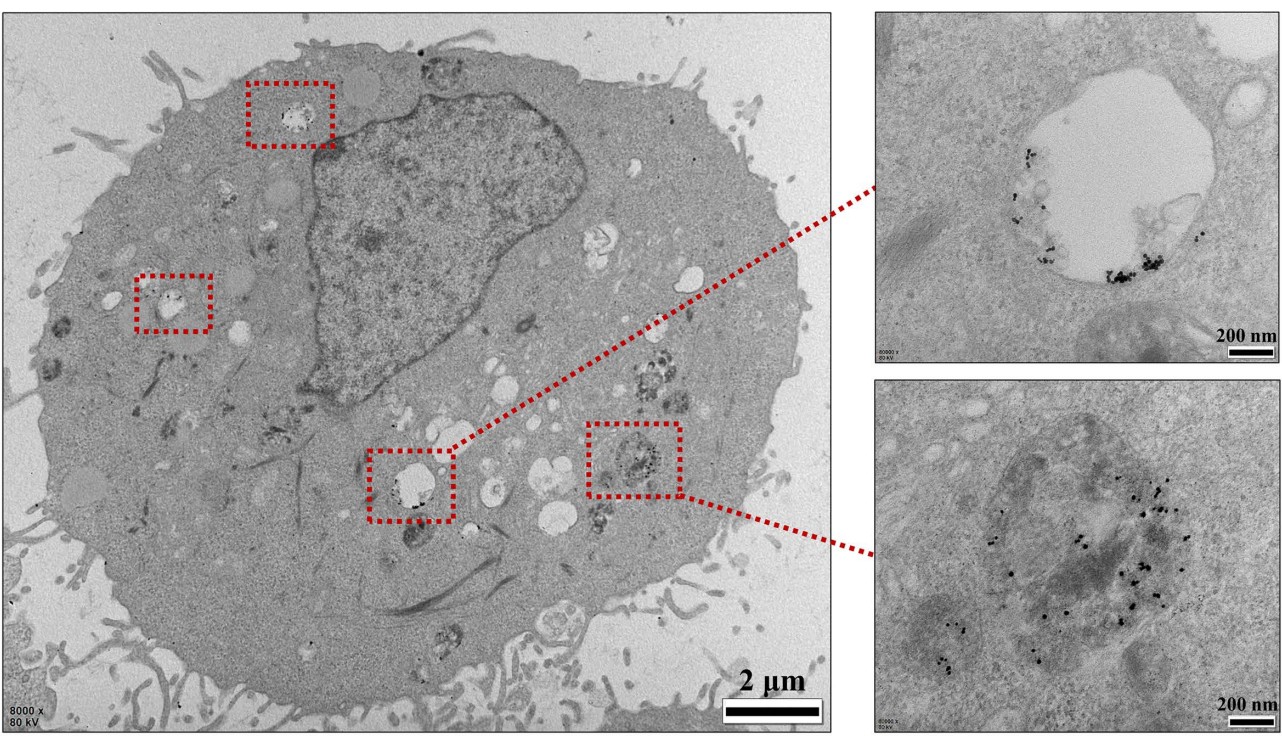

**Fig 5. TEM images of PHF@AuNPs distribution in A549 cells.** The particles are located in endo-lysosomes (Scale bar: 2 μm in the left image; 200 nm in the right images).

## Phototoxicity of PHF@AuNPs

Low cytotoxicity in dark and high phototoxicity were favourable for PDT. As we expected, PHF@AuNPs were biocompatible with A549 cells in a wide range we tested (< 50 μg/mL, Fig 7) because both AuNPs [31,32] and PHF [33,34] are considered as biologically safe particles. Cell viabilities of PHF, AuNPs, and PHF@AuNPs treated cells after 808 nm

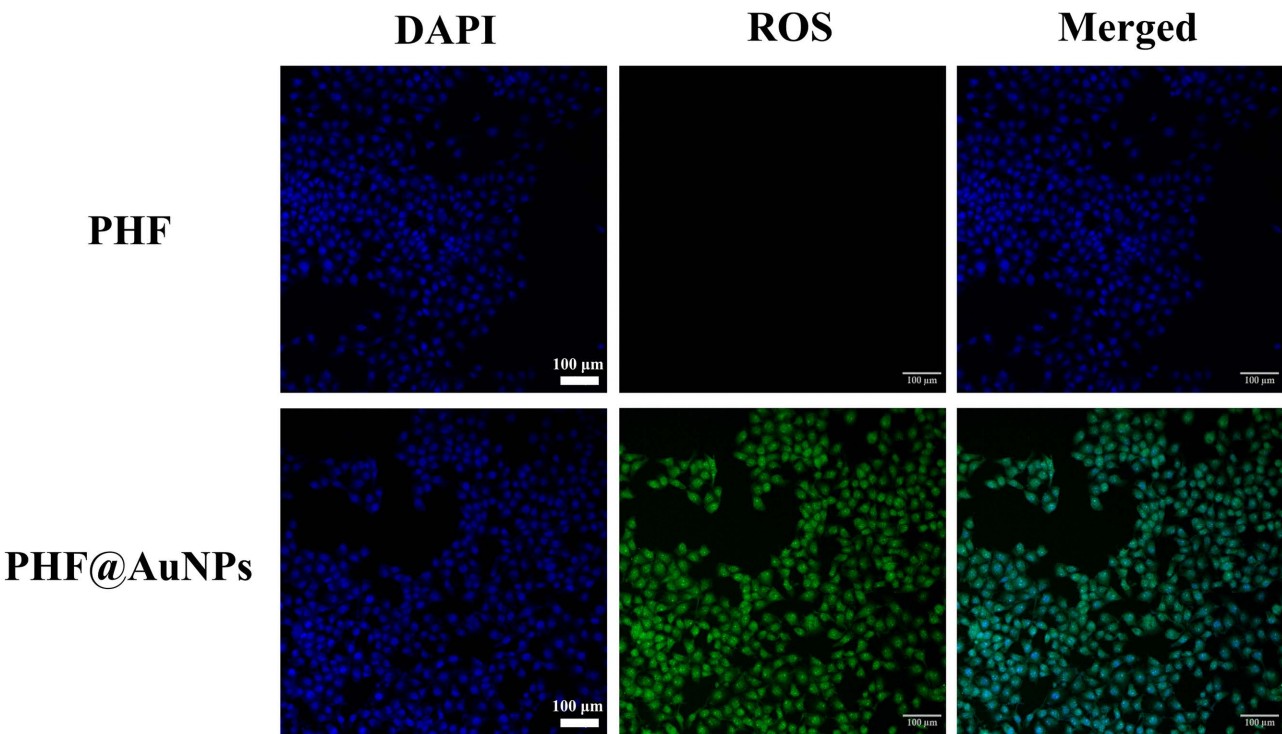

**Fig 6. Intracellular ROS detection in A549 cells.** Cells were treated with NIR laser after uptake of PHF@AuNPs (Scale bar: 100 μm).

treatment were shown in Fig 7. PHF@AuNPs exhibited concentration-dependent phototoxicity. The cell viability was 47.8±3.7% at the dose of 50 μg/mL. On the contrary, PHF and AuNPs hardly induced cell death after 808 nm laser irradiation. PHF couldn't be activated to yield ROS by NIR light as discussed above. Therefore, cells kept almost alive. Colloidal gold can absorb specific wavelength lights with high thermal conversion capacity attributed to its LSPR. Therefore, AuNPs have been extensively applied in photothermal therapy [35]. The absorption wavelength is positively related to particle size. For example, 22 nm AuNPs hold a maximum absorption spectrum at 517 nm, and 99 nm particles 580 nm. Therefore, AuNPs in the PHF@AuNPs complexes were not supposed to have photothermal activity upon 808 nm light [36]. We still set the AuNPs in phototoxicity study as the control to exclude the photothermal effect on cell viability. The results showed the obvious PDT efficacy of PHF@AuNPs.

## Conclusion

In total, we developed a novel nano-photosensitizer for NIR-triggered PDT treatment based on PHF@AuNPs. The composites were synthesized by a facile "mix and go" style reaction using, for the first time, PHF as a single reducing and capping agent. The features of PHF@AuNPs could be outlined as: (i) ultrafine size (~10 nm) with narrow size distribution; (ii) Core-shell structure with a PHF coating layer outside the composites; (iii) good biocompatibility, solubility and stability in physiological environments; (iv) efficient cellular uptake and potent photodynamic activity upon NIR irradiation.

This approach paves the way for engineering of fullerene based photosensitizers for NIR-triggered PDT. The *in vivo* study, including biodistribution and antitumor activity is needed to be performed for further investigation. The unique properties of the PHF@AuNPs suggest that, in perspective, it can be used as a new NIR light nano-photosensitizer for PDT treatment.

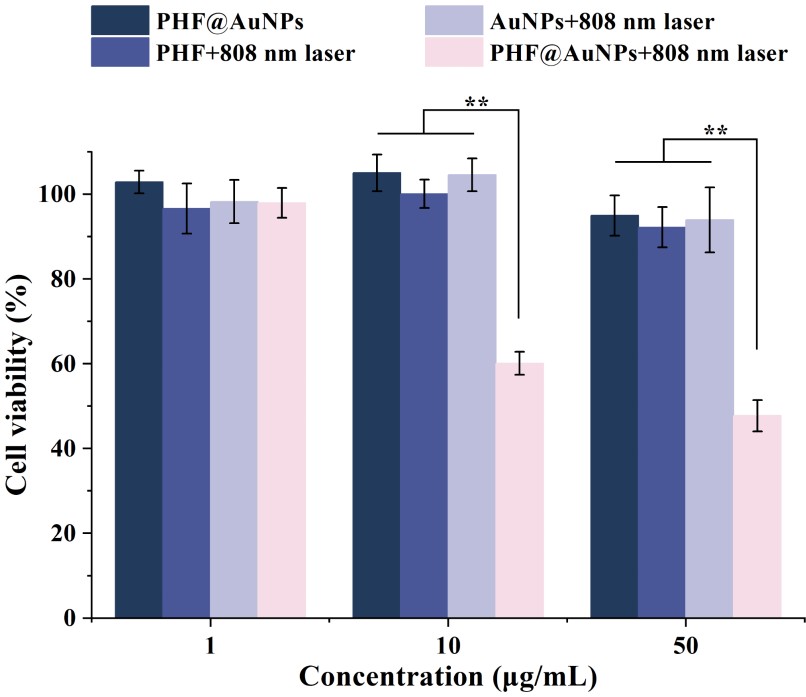

**Fig 7. Cell viability of A549 cells incubated with different concentrations of PHF@AuNPs, PHF, and AuNPs (1, 10, and 50 μg/mL) and subsequently irradiated by 808 nm laser for 10 min at power density of 1 W/cm² (\*\*p < 0.01, n = 6).**

## Supporting information

**S1 Fig. Schematic diagram of PHF@AuNPs synthesis, characterization and PDT activity evaluation.**
(TIF)

**S2 Fig. Size distribution of PHF@AuNPs in "a" and "b" group obtained at different reaction time measured by DLS.**
(TIF)

**S3 Fig. TEM characterization of PHF@AuNPs (A) obtained after 1 min reaction (Scale bar: 100 nm) and (B) with the un-optimized protocol (Scale bar: 200 nm).**
(TIF)

**S4 Fig. XPS analysis of PHF.**
(TIF)

**S5 Fig. EDS analysis of the chemical composition of PHF@AuNPs.**
(TIF)

## Author contributions

**Conceptualization:** Xiaoyi Sun, Yuanyuan Lv.

**Funding acquisition:** Yuanyuan Lv.

**Investigation:** Wenhao Li, Bohao Ruan, Xinyi Chen.

**Methodology:** Feng Zhou.

**Supervision:** Xiaoyi Sun, Yuanyuan Lv.

**Validation:** Xinyi Chen, Feng Zhou.

**Writing – original draft:** Wenhao Li, Bohao Ruan.

**Writing – review & editing:** Xiaoyi Sun.

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
