## [Decision Letter · Decision Letter 0]

11 Sep 2025

Dear Dr. Sun,

Thank you for submitting your manuscript to PLOS ONE. After careful consideration, we feel that it has merit but does not fully meet PLOS ONE’s publication criteria as it currently stands. Therefore, we invite you to submit a revised version of the manuscript that addresses the points raised during the review process.

We look forward to receiving your revised manuscript.

Kind regards,

Dinesh Kumar, Ph. D.

Academic Editor

PLOS ONE

Journal Requirements:

[Leading Goose R&D Program of Zhejiang - 2024C03230 - Yuanyuan Lv].

4. Thank you for stating the following in your manuscript:

[This study was supported by the Leading Goose R&D Program of Zhejiang (Grant nos. 2024C03230).]

[The author(s) received no specific funding for this work.]

6. PLOS requires an ORCID iD for the corresponding author in Editorial Manager on papers submitted after December 6th, 2016. Please ensure that you have an ORCID iD and that it is validated in Editorial Manager. To do this, go to ‘Update my Information’ (in the upper left-hand corner of the main menu), and click on the Fetch/Validate link next to the ORCID field. This will take you to the ORCID site and allow you to create a new iD or authenticate a pre-existing iD in Editorial Manager.

7. Thank you for stating the following in the Competing Interests section:

[The authors have declared that no competing interests exist.].   

We note that one or more of the authors are employed by a commercial company: Personalized Prescribing Inc.

8. Please upload a new copy of Figures 1, 3, 4, and 5 as the details are not clear. Please follow the link for more information: https://blogs.plos.org/plos/2019/06/looking-good-tips-for-creating-your-plos-figures-graphics/

Reviewers' comments:

Reviewer's Responses to Questions

**Comments to the Author**

1. Is the manuscript technically sound, and do the data support the conclusions?

Reviewer #1: Yes

Reviewer #2: Yes

2. Has the statistical analysis been performed appropriately and rigorously?

Reviewer #1: Yes

Reviewer #2: Yes

3. Have the authors made all data underlying the findings in their manuscript fully available?

Reviewer #1: Yes

Reviewer #2: Yes

4. Is the manuscript presented in an intelligible fashion and written in standard English?

Reviewer #1: Yes

Reviewer #2: Yes

Reviewer #1: 1. The abstract must be revised by addressing key results.

2. The first and second paragraphs of the introduction are too lengthy; they should be combined and shortened.

3. The description of Au NPs is not impressive; it should be revised by addressing the examples of such studies that are closer to your study. In the last paragraph of the introduction, you must discuss the gap in research, your study's novelty, and finally, the aim of the study.

4. Mention hours as "h" and minutes as "min," and mention the lyophilization conditions (time, temp, etc.) and model of lyophilizer.

5. Clearly describe the methodology of SEM, TEM, and DLS, and also mention the model of spectrometer.

6. ROS production by PHF@AuNPs: here don't use abbreviations at the start of the heading.

7. The sentence "contrasted with lead for TEM observation" is not clear.

8. The sentence "demonstrating the generation of AuNPs with similar particle size [23]" must be revised by mentioning the author of the study to which your AuNPs resemble.

9. The extended time of reaction resulted in a higher yield; for this statement, please discuss the mechanism with supportive literature.

10. In Fig. 1D and 1E, write the numerical values at their peaks. While in 1 C (a, b, c) and 1 F, the scale is missing. In Fig 4 and Fig 5 scale bar and value are not clear.

11. Results of Fig 4 and Fig 5 should be discussed with literature.

12. Revise the conclusion.

Reviewer #2: This manuscript presents a novel and facile synthesis of PHF@AuNPs nanocomposites for effective NIR-triggered photodynamic therapy, demonstrating promising in vitro results and stability. The methodology and discussion are generally clear. However, this manuscript requires improvement to enhance clarity, data presentation, and scientific rigor.

1. The abstract succinctly introduces the research and highlights key findings. However, it should clarify experimental conditions and include quantitative results to enhance its impact.

2. The introduction provides relevant background and highlights the importance of NIR-triggered photodynamic therapy. However, several statements require clearer citations and a more critical comparison to previous research. For example, when describing the limitations of existing photosensitizers and their absorption wavelengths, the manuscript should directly cite recent key studies or reviews. Additionally, the claims about the superiority of PHF@AuNPs should be critically compared with previous nanocomposite systems by referencing published performance metrics or outcomes.

3. The methods section describes procedural details sufficiently, but it lacks clarity in experimental design and reproducibility. Concentrations, incubation times, and analytical techniques should be presented in a more structured and concise manner. Additionally, a flow diagram or schematic could greatly improve readers’ understanding of the synthesis and characterization workflow.

4. The results and discussion are comprehensive and present clear experimental findings regarding the synthesis, characterization, and biological evaluation of PHF@AuNPs nanocomposites. However, the section would benefit from additional quantitative data, clearer comparative tables, and improved image quality. Authors should provide more critical analysis by contrasting their findings with previous studies and elaborate mechanisms underlying the observed effects.

5. The conclusion concisely summarizes the study’s achievements but should more clearly state the broader significance and limitations.

**Do you want your identity to be public for this peer review?** For information about this choice, including consent withdrawal, please see our Privacy Policy

Reviewer #1: No

Reviewer #2: No

---

## [Author Response · Author response to Decision Letter 1]

25 Oct 2025

Dear Editor,

Thank you very much for your letter and comments from the academic editor and reviewers about our paper submitted to PLOS ONE (Manuscript ID: PONE-D-25-44370).

We amended the Funding Statement as following: Yuanyuan Lv: Leading Goose R&D Program of Zhejiang (Grant nos. 2024C03230). https://kjt.zj.gov.cn. The funder provide the fund to support manuscript for publication. The funder had no role in study design, data collection and analysis, decision to publish, or preparation of the manuscript. There was no additional external funding received for this study.

The Competing Interests Statement is uploaded as following: The authors have declared that no competing interests exist. The commercial affiliation (Personalized Prescribing Inc.) does not alter our adherence to all PLOS ONE policies on sharing data and materials.

We have checked the manuscript and revised it according to the comments. We upload the separate file labeled “Response to Reviews”, “Revised Manuscript with Track Change” and an unmarked version of manuscript without tracked changes. . If you have any question about this paper, please don’t hesitate to let us know. 

Sincerely yours,

Xiaoyi Sun

---

## [Editor Report · Decision Letter 1]

29 Oct 2025

One Step Synthesis of Ultrafine PHF@AuNPs Nanocomposite and its Application in NIR Triggered Photodynamic Therapy

PONE-D-25-44370R1

Dear Dr. Yuanyuan Lv,

We’re pleased to inform you that your manuscript has been judged scientifically suitable for publication and will be formally accepted for publication once it meets all outstanding technical requirements.

Kind regards,

Dinesh Kumar, Ph. D.

Academic Editor

PLOS ONE
---

## [Editor Report · Acceptance letter]

PONE-D-25-44370R1

PLOS ONE

Dear Dr. Sun,

I'm pleased to inform you that your manuscript has been deemed suitable for publication in PLOS ONE. Congratulations! Your manuscript is now being handed over to our production team.

Kind regards,

on behalf of

Dr. Dinesh Kumar

Academic Editor

PLOS ONE